# A New Species *Agrocybe striatipes*, also a Newly Commercially Cultivated Mushroom with Highly Nutritional and Healthy Values

**DOI:** 10.3390/jof9030383

**Published:** 2023-03-21

**Authors:** Jiaxin Li, Wenqiang Yang, Jinwei Ren, Bin Cao, Xinyu Zhu, Li Lin, Wen Ye, Ruilin Zhao

**Affiliations:** 1State Key Laboratory of Mycology, Institute of Microbiology, Chinese Academy of Sciences, Beijing 100101, China; 2College of Life Sciences, University of Chinese Academy of Sciences, Beijing 100049, China; 3College of Science, Gansu Agricultural University, Lanzhou 730070, China; 4Center of Excellence in Fungal Research, Mae Fah Luang University, Chiang Rai 57100, Thailand; 5Xianheyi Agricultural Technology Development Co., Yibin 644400, China; 6Huayuan Bank Ecological Agricultural Development Co., Yibin 644400, China

**Keywords:** edible mushrooms, nutritional components, phylogeny

## Abstract

The species of *Agrocybe* (Strophariaceae, Agaricales, Agaricomycetes) are saprophytic and widely distributed in temperate regions. In this study, a new species named *Agrocybe striatipes* from China is described, which has been successfully cultivated in China recently. The phenotypic characteristics examination and molecular phylogenetic analyses using multilocus data (ITS and nrLSU) both support it as a new species in the genus *Agrocybe*. Moreover, nutritional ingredient analysis showed that the fruiting body of *A. striatipes* was rich in seventeen amino acids, including eight essential amino acids, in addition to high levels of calcium (78.5 mg/kg) and vitamin D (44.1 μg/100g). The following analysis of the heavy metal contents of the fruiting bodies show that it does not contain lead, cadmium, arsenic, mercury, and other heavy metal elements. In the crude extract of the mushroom, the nutrients in the aqueous phase are amino acids and oligosaccharides, and the active substances in the ethyl acetate layer are sterols, which have a variety of pharmacological effects. In conclusion, *A. striatipes* is not only a new species but also has highly application values as a cultivated edible mushroom in nutrition and health.

## 1. Introduction

The genus *Agrocybe* was first named by Fayod in 1889 with the type species *A. praecox* (Pers.) Fayod. *Agrocybe* species are highly adaptable and widely distributed in temperate regions of Asia, Europe, and North America [1]. In China, sixteen species/varieties were reported from the Guizhou, Yunnan, Fujian, Sichuan, Jiangsu, Zhejiang provinces, etc. [2,3]. Recently, the outline of Basidiomycota documented c. 100 species in this genus [4]. Members of *Agrocybe* are saprophytic, usually grow in the forest or grassland, and are characterized by small-to-medium-sized basidiomata, most of them have a membranous ring, brown spore prints, pileipellis a hymeniderm or ixohymeniderm, basidiospores ovoid, ellipsoid or fusiform, yellow to light brown, smooth, usually with a broad germ-pore [5,6].

Singer conducted a comprehensive morphological classification study on *Agrocybe,* and placed this genus in the family Bolbitiaceae based on characteristics of the pileipellis cell consisting of pear-shaped, subspherical, sphaerocytes cells; the presence or absence of hymenial cystidia; and brown spore prints [7]. Furthermore, he divided this genus into two subgenera, subg. *Agrocybe* and *Aporus*; subg. *Agrocybe* was further divided into five sections (sections *Agrocybe*, *Pediades*, *Microsporae*, *Allocystides*, and *Evelatae*), and subg. *Aporus* was separated into two sections (sections *Aporus*, *Velatae*) [7]. Later, the section *Evelatae* was moved into subg. *Aporus* from subg. *Agrocybe*, while section the *Allocystides* was deleted from the taxonomic system of this genus [5]. This classification system has been used widely [2,5,8].

However, the molecular phylogenetic studies indicated that the genus *Agrocybe* should belong to the family Strophariaceae (Agaricales, Agaricomycetes) [9], which has been accepted in the modern Basidiomycota taxonomic system [4,10]. Furthermore, the molecular phylogenetic analyses that used ITS and nrLSU sequence revealed that the genus *Agrocybe* was a polyphyletic group composed of four clades [11,12]. The species from three of those clades were kept in *Agrocybe*, including the type species *A. praecox* (Pers.) Fayod. In contrast, the clade contained the widely cultivated the species “Chashugu” *Cyclocybe cylindracea* (=*A. cylindracea*), *C. erebia* (=*A. erebia*), and *C. erebioides* (=*A. erebioides*) formerly from subg. *Aporus* that were moved to the genus *Cyclocybe* Velen. [12,13]. More recently, another phylogenetic research revealed one more clade from the genus *Agrocybe.* Therefore, *Agrocybe* contained four clades total, but still was a polyphyletic group [8].

It is well known that the genus *Cyclocybe*, which is similar to *Agrocybe* in morphology and phylogeny, contains many edible species, and some of them have been successfully cultivated, including *C. cylindracea* (DC.) Vizzini & Angelini, *C. salicaceicola* (Zhu L. Yang, M. Zang & X.X. Liu) Vizzini, and *C. chaxingu* (N.L. Huang) Q.M. Liu, Yang Gao & D.M. Hu [3,12,14]. These species are popular because of their high nutritional properties such as high protein, low fat, and having bioactive ingredients with potential pharmacological effects such as antioxidant and anti-aging properties [13,15]. In recent years, *A. praecox*, the type species of *Agrocybe,* has been successfully cultivated [16], and presently it is the only species of this genus that has been domesticated. The following analysis of the cultivated fruiting bodies of *A. praecox* show that it contains rich essential amino acids and K (2190mg/kg) [16]. Therefore it should be possible that there are more species from this genus that could be explored as new food resources.

In this study, the specimens of *Agrocybe* were collected from Sichuan Province, China, and the morphological and molecular phylogenetic analysis showed that it represented a new species. In addition, the artificial domestic cultivation of this species was successfully carried out. The analyses of nutrients and bioactive ingredients show that this mushroom is rich in amino acids, elemental calcium, vitamin D, and ergosterol. We concluded that this species could be a great cultivated mushroom for commercial development.

## 2. Materials and Methods

### 2.1. Materials

Mushroom specimens were collected from 28.36 N, 105.12 E, FuAn Village, Bo Wang Shan Town, Xingwen County, Yibin City, Sichuan Province, China, and deposited in the Herbarium Mycologicum Academiae Sinicae, Beijing, China (HMAS). Strains were isolated from fresh fruiting bodies and deposited in the China General Microbial Strain Collection Management Center, Conservation number CGMCC 40360.

### 2.2. Morphological Study

Mushroom specimens were collected in the field after taking photographs. Odor and color changes on bruising were recorded at the same time. Macromorphological features and chemical reactions of fresh specimens were recorded. Specimens were dried completely with a food drier under a temperature of 55 °C overnight. Anatomical and cytological features including lamellae, pileipellis, basidiospores, basidia, and cystidia were observed from dried specimens and following the protocols [17,18,19]. A total of 5% KOH were used for a staining reaction. More than twenty measurements of microscopic features (spores, basidia, and cystidia) were recorded, which included tx, the mean of the length by the width ± SD; Q, the quotient of the basidiospore length to width; and Qm, the mean of the Q-values ± SD [17,20].

### 2.3. Molecular Phylogenetic Study

DNA was extracted from the dried specimens using a Broad-spectrum plant Rapid Genomic DNA Kit (Biomed) according to the manufacturer’s protocol. Primers ITS4 and ITS5 were used for internal transcribed spacer (ITS) and LROR and LR5 for large ribosomal subunit (nrLSU) PCR reactions [21]. The PCR programs followed previous studies [19,20,22]. The PCR products were sent to a Biomed Biotechnology commercial company for sequencing.

The sequences produced from this study and some from the NCBI GenBank database were used in phylogenetic analyses [3,8,12,13,23,24] (Table 1). Sequences of multigene data were aligned by Muscle version 3.6 separately [25], then manually adjusted to remove ambiguous regions in BioEdit version 7.0.4 [26]. Maximum likelihood (ML) analysis was performed by RAxmlGUI 1.3 under a GTRGAMMA model with one thousand rapid bootstrap (BS) replicates [27]. Bayesian Inference (BI) analysis was performed by MrBayes v3.2.6 [28]. Six Markov chains were run for 2,000,000 generations and trees were sampled every 100th generation. Burn-ins were determined in Tracer version 1.6 with an ESS value higher than 200, and the remaining trees were used to calculate Bayesian posterior probabilities (PP). The trees were displayed in FigTree version 1.4.0 [29].

### 2.4. Nutritional Analysis of Fruiting Bodies

#### 2.4.1. Nutritional Composition Analysis

Nutrient composition analysis was performed using a completely dried artificially cultivated fruiting body. The amino acid content, polysaccharide content, protein content, ash content, mineral content, fat content, energy, and carbohydrate were determined by sending it to the Analysis and Testing Center of Sichuan Academy of Agricultural Sciences. An automatic amino acid analyzer (L-8800) and atomic-fluorescence photometer (AFS3000) were used to assay the amino acid and mineral contents. All determinations are under China’s National Food Safety Standard System, respectively [30,31,32,33,34,35]. A comparative analysis of the nutrient compositions of this proposed new species and eight reported cultivated species, namely *A. praecox*, *C. cylindracea*, *C. salicacicola*, *Pleurotus placentodes* (Rumph. ex Fr.) Boedijn, P. ostreatus (Jacq.) P. Kumm., *Lentinus edode* (Berk.) Pegler, *Flammulina velutipes* (Curtis) Singer, and *Agaricus bisporus* (J.E. Lange) Imbach, was conducted.

#### 2.4.2. Heavy Metal Content Analysis

The artificially cultivated fruiting body was detected for heavy metal content. Pb (Lead), Cd (Cadmium), Hg (Mercury), and As (Arsenic) were determined by sending them to the Analysis and Testing Center of Sichuan Academy of Agricultural Sciences. An atomic fluorescence photometer (AFS3000) was used in the analysis based on the atomic fluorescence spectrometric method, and all determinations were performed according to the corresponding China National Food Safety Standard [36].

#### 2.4.3. Metabolites Analysis

The target compound was purified by HPLC (Agilent 1200), which was performed on C_18_ column (4.6 mm × 250 mm, 5 μm) with a flow rate of 1.0 mL/min. The mobile phase consisted of methanol (A) and water containing 0.01% formic acid (B). A gradient elution program was set as follows: 0–5 min, 5% A; 5–40 min, 5–100% A. The sample injection volume was 10 μL and the column temperature was set at 25 °C. Its structure was determined by ^1^H and ^13^C NMR data analysis.

## 3. Results

### 3.1. Phylogenetic Results

Ninety-five samples were included in this multigene phylogenetic analysis, and it contained 50 *Agrocybe* samples representing 23 species; the genus *Cyclocybe* was set as the outgroup. The overall topologies of the ML and BI trees did not show a difference, and the ML tree is shown in Figure 1. Among them, six ITS sequences and six nrLSU sequences were newly generated for this study. It is noticeable that *Agrocybe* was not a monophy-letic group, and all *Agrocybe* species split into four clades, which correspond with a previous study by Kiyashko [8]. The proposed new species *A. striatipes* was nested in the Clade I, which comprised the type species of this genus, and it also showed its sister relationship with the known species *A. smithii* Watling & H.E. Bigelow, then clustered with the other known *Agrocybe* species (*A. firma* (Peck) Singer, *A. acericola* (Peck) Singer, *A. praecox*, and *A. pediades* (Fr.) Fayod). The new species is represented by specimens (HMAS 286942, HMAS 286943) and strains (JL2022301, JL2022302, CGMCC 40360, ZRLJL2021002) which form a distinct lineage with strong supported values (99 BS/1 PP).

### 3.2. Taxonomy

*Agrocybe striatipes* R.L. Zhao & J.X. Li, sp. nov. Figure 2 and Figure 3

Fungal Names number: FN571233

Etymology: referring to the surface of stem striate.

Holotype: HMAS286942 (*ZRL20211296*). China, Sichuan Province, Yibin City, Xingwen County, Bo Wang Shan Town, FuAn Village, 28.36 N, 105.12 E, 28 July 2021, collected by Rui-Lin Zhao.

Macroscopic description: Pileus 18–45 mm diameter, light yellowish, ochraceous-brown, darkening in the center, sometimes completely white in cultivation, obtusely umbonate with a distinct papilla, surface dry, smooth, usually with irregular rugose when mature, margin occasionally decurved. Lamellae-free, adnate-to-sub-decurrent, dark-brown, chestnut color, crowded. Stipe 65–125 × 30–80 mm, fibrillose, striate-sulcate, cylindrical, hollow, equal except for the enlarged base, with abundant white rhizomorphs. Annulus absent. Spore-prints rusty-brown to ocher. Context white-to-ochraceous, up to 2 mm thick at the disk.

Microscopic description: Basidiospores 8.1–10 × 5.2–6.9 μm, [X = 8.7 ± 0.4 × 6.2 ± 0.4, Q = 1.2–1.7, Qm = 1.4 ± 0.1, *n* = 22], ellipsoid, occasionally globose to subglobose, smooth, reddish-brown, thick-walled, apically truncate with a germ-pore, up to 1–2 μm. Basidia 23.6–31.1 × 8–10.5 μm, clavate, hyaline, four-spored, occasionally three-spored. Pleurocystidia 34–47.2 × 24.7–28 μm, ventricose with rounded obstuse apex, thin-walled, hyaline. Cheilocystidia 38.4–46.1 × 20–27.9 μm, similar to pleurocystidia, ventricose with a broadly rounded apex. Pileipellis hymeniform, composed of calvate, hyaline hyphae, 1.1–7.6 μm, with vesicular-clavate cells at the apex, 15.6–58.6 × 9–15 μm. Stipitipellis composed of branched, calvate, hyaline hyphae, thick-walled, 3.4–10 μm. Clamp-connections abundant.

KOH reaction: not distinctive

Habit: gregarious on soil.

Known distribution: Sichuan Province, China.

Other examined materials: HMAS486943 (*ZRL20220014*). China, Sichuan Province, Yibin City, Xingwen County, Bo Wang Shan Town, FuAn Village, 28.37 N, 105.09 E, 21 May 2022, collected by Rui-Lin Zhao.

Notes: The new species *A. striatipes* is characterized by its yellowish ochraceous-brown pileus, stipe deeply striate-sulcate with somewhat fibril, relatively larger pleurocystidia and cheilocystidia, and smaller basidiospores than other species. In the phylogenetic tree (Figure 1), *A. striatipes* and *A. smithii* formed a distinct lineage and separated from the other *Agrocybe* species. However, *A. smithii* can be easily distinguished by a fresh cap with olive shades and having bigger basidiospores (11–13.5 × 6.5–8 μm) [37]. *A. allocystis* and *A. striatipes* are very similar in the field in terms of morphological characteristics; however, the former has much bigger basidiospores (10–16 × 7–10.5 μm) and is different in pleurocystidia and cheilocystidia shapes, which were ventricose to lageniform, and apically usually sub capitate to capitate [6]. Another morphologically similar species is *A. broadwayi*, which differs from *A. striatipes* by its bigger basidiospores (12–15 × 8–9 μm) and margin of pileus usually striated, occasionally covered with small concolorous squamules [38]. *A. retigera* resembles *A. striatipes* too because they both have umbonate, yellow-cream colored to pale brownish pileus surfaces; however, this known species’ heavy lacunose-rugose will disappear in older ones, and it possesses bigger basidiospores (11.5–18 × 7–10 μm) with a double wall [39].

There are pure-white fruiting bodies that occurred in the cultivated yard and mixed with those in brown. Due to the fact that they have the same morphological characters except for the color and identical sequences, they were identified as the same species. In *Agaricus* and some other mushroom genera, these white variants are very frequent in many species, such as *A. bisporus*, *A. subrufescens* [40,41].

### 3.3. Evaluations of Nutrition and Food Security of A. striatipes

The nutrient content of *A. striatipes* was analyzed from its dry materials, and the details are shown in Table 2. The protein content of *A. striatipes* was 5.66 g/100 g, and its content was nearly three times as much as those of *A. praecox*. The richness of calcium (78.5 mg/kg) and zinc (6.87 mg/100 g) is beneficial to the elderly and growing children; in addition, trace amounts of selenium have been detected, which is beneficial to the normal physiological activities of the human body. Furthermore, the cultivated specimens were also rich in vitamins, with a vitamin D content of 44.1 μg/100 g. Vitamin C and other fat-soluble vitamins were not detected within the limit of quantification.

As shown in Table 3, a comparative analysis of amino acid content with another eight common edible mushrooms was conducted. Our analysis revealed that this species is rich in 17 amino acids; the total amino acid is 19.24 g/100 g, including the eight essential amino acids required by the human body. In addition, we have detected gamma-aminobutyric acid, which was not reported in others. The ratios of essential and non-essential amino acids indicated that these two were in equal amounts.

The analysis results of heavy metal contents are detailed in Table 4, and the results show that the artificially cultivated fruiting body did not contain lead, cadmium, arsenic, mercury, or other heavy metal elements, and their contents were within the standard limits.

### 3.4. The Bioactive Ingredients of A. striatipes

The dry encarpium (200 g) was extracted repeatedly with 95% ethanol 500 mL three times, and the organic solvent was evaporated to dryness under a vacuum to afford the crude extract (3.2 g), which was distributed between water and EtOAc to afford the two fractions. They were analyzed by HPLC. The water fraction was dissolved in DMSO-d6 and analyzed by ^1^H NMR, which revealed that the water fraction mainly contained oligosaccharides and amino acids (Appendix A Figure A1). We found that there was a main compound with a retention time of 40.0 min; other peaks with similar retention times have the same UV absorption, which were as shown in Figure 4. The main compound (3.0 mg) was purified by the same HPLC condition. It was identified as ergosterol by an analysis of ^1^H and ^13^C NMR contrast the reference (Appendix B Figure A2 and Appendix C Figure A3) [42]. Ergosterol, the major product of mycosterol biosynthesis, is an important component of fungal cell membranes that maintain membrane structural integrity, permeability, and fluidity, which can promote the absorption of calcium and phosphorus in the human body. It is one of the good sources of exogenous vitamin D2 in the human body and is often used to improve rickets, osteomalacia, and osteoporosis caused by vitamin D deficiency in infants and elderly people [43].

## 4. Discussion

Although species of the generas *Agrocybe* and *Cyclocybe* are morphologically similar, the molecular phylogenetic analysis revealed that *Cyclocybe*, which is represented by the widely cultivated *C. cylindracea* complex (=*A. cylindracea*), belongs to Tubariaceae. However, the phylogenetic study that used multigene sequences of Strophariaceae, Tubariaceae, and other selected Hymenogastraceae species showed that *Agrocybe* are in fact phylogenetically distant from *Cyclocybe* and belong to Strophariaceae [24]. In morphology, species of *Agrocybe* possess a broad germ-pore; however, *Cyclocybe* species rarely possess germ-pore [6,12].

In this study, a multi-gene phylogenetic analysis was carried out based on 95 specimens, including 50 specimens of 23 *Agrocybe* species. Phylogenetic results enable the division of the current *Agrocybe* into four main clades, which coincided with previous research [8,12,13]. Clade I contains the proposed new species and the type species *A. praecox*, as well as most of the *Agrocybe* species. Clade II contains only species *A. arvalis*, which is nested at the sister position with Clade I without statistics supporting the values. The Clade IV is located at the base of the phylogenetic tree and comprises five strongly supported species (97 BS/1 PP). Regardless, the molecular analysis supports *A. striatipes* as a new species, and this is also supported by its morphological characteristics. To resolve the taxonomic problem of polyphyletic *Agrocybe*, the taxonomic system for those related taxa may change in the future. However, no matter how the taxonomic system changes, *A. striatipes* as belonging to *Agrocybe* will not change due to it being closely clustered with type species *A. praecox* within clade I in the phylogeny (Figure 1) [8].

Currently, eight edible species of Agrocybe were reported in China [44], and only *A. praecox* was successfully domesticated and cultivated. The new species *A. striatipes* introduced from this study is the second species of *Agrocybe* that can be capable of cultivation. The results of the nutrient analysis revealed that the cultivated *A. striatipes* can be used as a nutritious food high in proteins. As shown in Table 2, the protein content of *A. striatipes* is about 5.66 g/100 g, which is significantly higher than that of *A. praecox*. In addition, it is rich in polysaccharides. Further analysis of polysaccharide extraction and identification of components can be carried out to develop the medicinal value of its polysaccharide active substances and other components. *A. striatipes* is also rich in mineral elements, with a calcium content (78.5 mg/kg) significantly higher than that of *A. praecox*. Meanwhile, *A. striatipes* substrates are rich in Vitamin B2 (0.33 mg/100 g), and B vitamins are mainly involved in bio-oxidation and metabolism in the form of coenzymes, which have very important physiological functions. From the results of the amino acid content, *A. striatipes* is rich in amino acids, with a total amino acid content of 4.14 g/100 g, which is significantly higher than that of *A. praecox*, and GLU is the highest amino acid content, accounting for 18%. However, compared with other common edible mushrooms, the total amino acids of *A. striatipes* are not high, but the ratio of essential amino acids to total amino acids is comparable to other edible mushroom species. 

In the crude extract of the mushroom, the nutrients in the aqueous phase are amino acids and oligosaccharides, and the active substances in the ethyl acetate layer are sterols. Ergosterol is the most common class of active substances in fungi. Ergosterol has a variety of pharmacological effects, such as strengthening the immune system, anti-inflammatory and pain-relief properties, lowering cholesterol, anti-fibrosis, anti-oxidant, and delaying aging [45,46].

## 5. Conclusions

This study, which involved multiple DNA gene-fragment analyses in combination with morphological analysis, revealed that *A. striatipes* is highly supported as a new species. The success of its artificial cultivation proved that it was the second species in the genus that can be successfully cultivated. Furthermore, the evaluations of its nutrition, food security, and bioactive ingredients indicated that it could be a healthy food for human beings. Compared with *A. praecox*, *A. striatipes* produces higher levels of proteins and amino acids and is richer in calcium, zinc, and vitamin D contents, which especially make it more suitable for the elderly and growing children. In the daily diet structure, we recommend combining *A. striatipes* with other edible mushrooms, vegetables, or meats, which can provide a complementary balance of amino acids and mineral elements.

## Figures and Tables

**Figure 1 jof-09-00383-f001:**
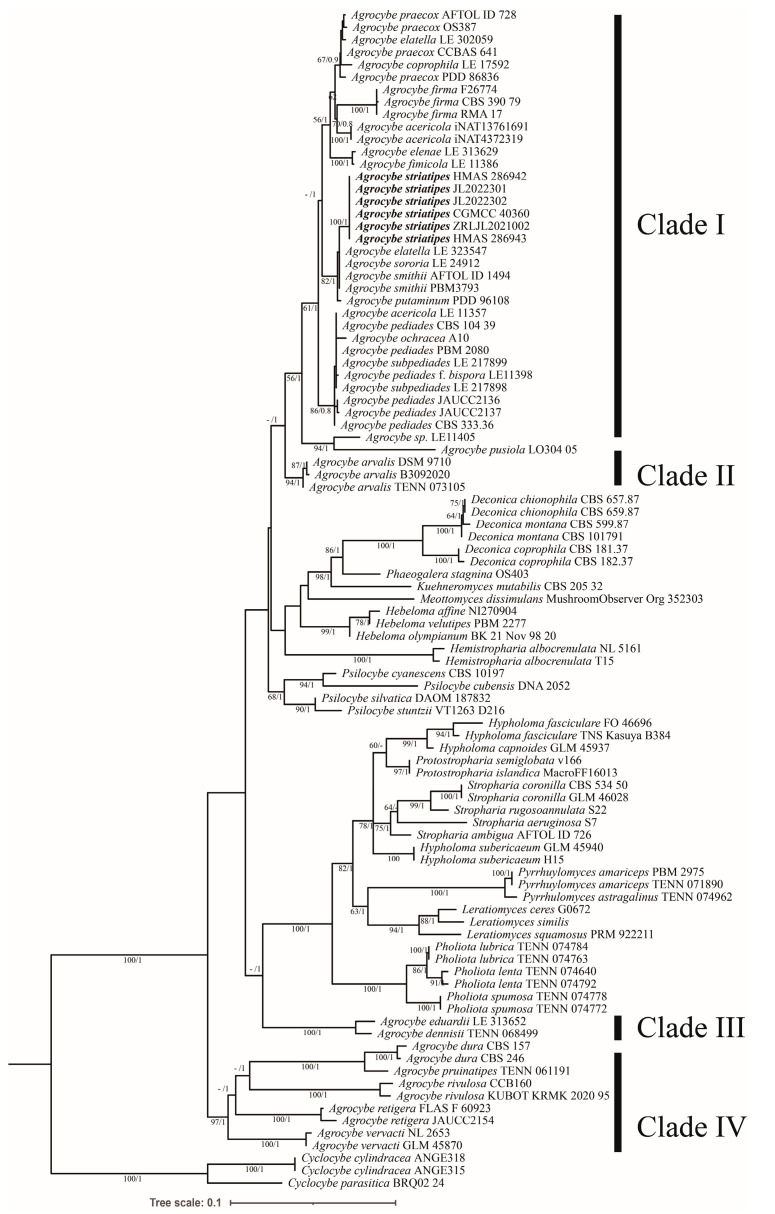
Maximum likelihood (ML) tree of *Agrocybe* based on ITS and nrLSU sequences data, including species selected from Hymenogastraceae and Strophariaceae, rooted with *Cyclocybe*. The bootstrap values and Bayesian posterior probabilities of more than 50%/0.8 (BS/PP) are indicated at the nodes.

**Figure 2 jof-09-00383-f002:**
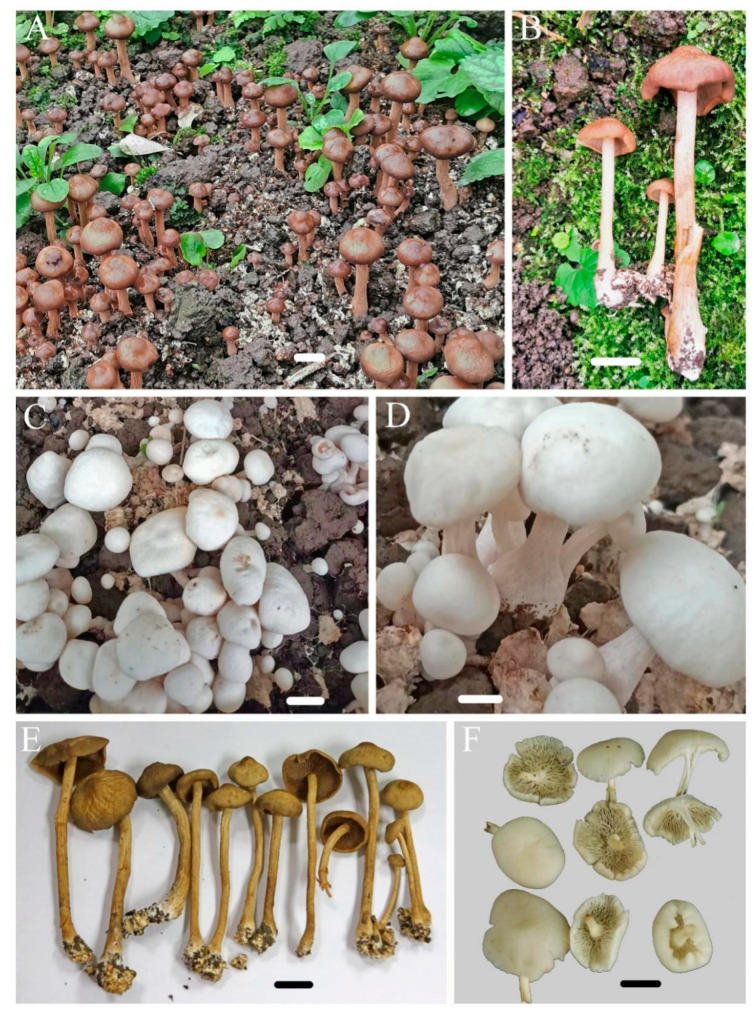
The macromorphological characters of *Agrocybe striatipes* in cultivation. (**A**,**B**,**E**) holotype (HMAS286942); (**C**,**D**,**F**) specimen HMAS286943 with pure-white fruiting bodies. Scale bar: (**A**–**C**) = 2 cm; (**D**) = 1 cm, (**E**,**F**) = 2 cm.

**Figure 3 jof-09-00383-f003:**
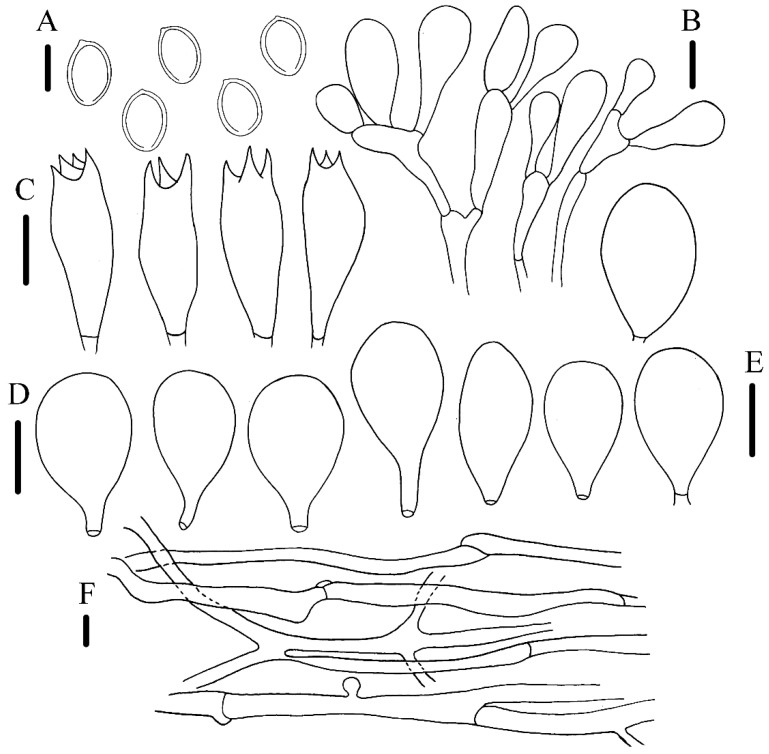
Micromorphology of *A. striatipes* (HMAS486942, holotype). (**A**) basidiospores; (**B**) pileipellis hyphae; (**C**) basidia; (**D**,**E**) cheilocystidia and pleurocystidia; (**F**) stipitipellis hyphae. Scale bar: (**A**–**F**) = 5 μm.

**Figure 4 jof-09-00383-f004:**
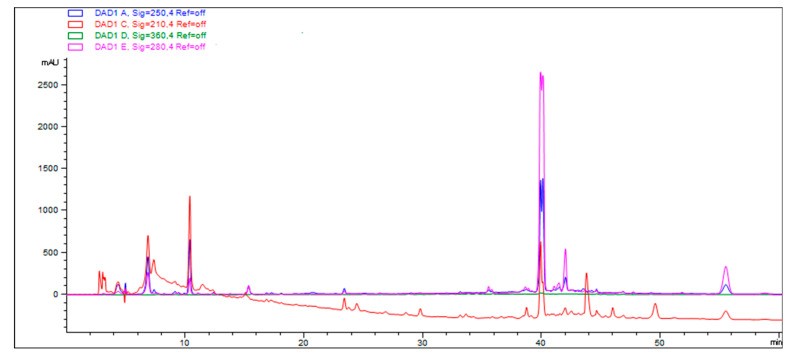
Analysis of the main compound of water and EtOAc fractions by HPLC.

**Table 1 jof-09-00383-t001:** Sequence information of *Agrocybe* used in molecular phylogenetic analyses; newly generated sequences are in red. Missing sequences are indicated by “-”.

Species	Voucher Number/Strains	Location	GB Accession Numbers	
			ITS	nrLSU
*Agrocybe acericola*	iNAT4372319	USA	MZ158314	-
*Agrocybe acericola*	iNAT13761691	USA	MZ158565	-
*Agrocybe acericola*	LE 11357	Kyrgyzstan	JN684805	-
*Agrocybe arvalis*	DSM 9710	Germany	MN306191	MN306170
*Agrocybe arvalis*	B3092020	USA	MW349111	-
*Agrocybe arvalis*	TENN 073105	USA	MH615058	MT237464
*Agrocybe coprophila*	LE 17592	Russia	OM524384	OM523960
*Agrocybe dennisii*	TENN 068499	USA	KY744153	MF797665
*Agrocybe dura*	CBS 157	Unknown	MH858248	MH869851
*Agrocybe dura*	CBS 246	Unknown	MH855957	MH867453
*Agrocybe eduardii*	LE 313652	Russia	OM524381	OM523957
*Agrocybe elatella*	LE 302059	Russia	OM524385	OM523961
*Agrocybe elatella*	LE 323547	Russia	OM524386	OM523962
*Agrocybe elenae*	LE 313629	Russia	OM524382	OM523958
*Agrocybe fimicola*	LE 11386	Russia	OM524383	OM523959
*Agrocybe firma*	F26774	USA	MZ314314	-
*Agrocybe firma*	CBS 390.79	Unknown	MN306192	MN306171
*Agrocybe firma*	RMA 17	USA	MG663239	MT237458
*Agrocybe striatipes*	ZRL20211296	China	OQ186168	OQ186162
*Agrocybe striatipes*	ZRL20220014	China	OQ186171	OQ186165
*Agrocybe striatipes*	ZRLJL2021001	China	OQ186172	OQ186166
*Agrocybe striatipes*	ZRLJL2021002	China	OQ186173	OQ186167
*Agrocybe striatipes*	JL2022301	China	OQ186169	OQ186163
*Agrocybe striatipes*	JL2022302	China	OQ186170	OQ186164
*Agrocybe ochracea*	A10	India	MG383657	-
*Agrocybe pediades*	JAUCC2136	China	MN715758	MN710538
*Agrocybe pediades*	JAUCC2137	China	MN715759	MN710539
*Agrocybe pediades*	PBM 2080	-	DQ484057	-
*Agrocybe pediades*	CBS 104.39	-	MH855969	MH867465
*Agrocybe pediades*	CBS 333.36	-	-	MH877770
*Agrocybe pediades f. bispora*	LE11398	Russia	JN684774	-
*Agrocybe praecox*	AFTOL_ID 728	Unknown	AY818348	AY646101
*Agrocybe praecox*	OS387	Norway	KC842389	KC842460
*Agrocybe praecox*	PDD 86836	New Zealand	KM975410	KM975356
*Agrocybe praecox*	CCBAS 641	Czech Republic	MN530062	MN528792
*Agrocybe pruinatipes*	TENN 061191	USA	OM523930	-
*Agrocybe pusiola*	LO304_05	-	DQ389732	-
*Agrocybe putaminum*	PDD 96108	New Zealand	KM975434	KM975371
*Agrocybe retigera*	JAUCC2154	China	MT755839	MN710544
*Agrocybe retigera*	FLAS_F_60923	USA	MH016951	MH620258
*Agrocybe rivulosa*	CCB160	USA	KF830098	KF830090
*Agrocybe rivulosa*	KUBOT_KRMK_2020_95	India	MW487609	MW485813
*Agrocybe smithii*	AFTOL_ID 1494	Unknown	DQ484058	DQ110873
*Agrocybe smithii*	PBM3793	USA	MG663269	-
*Agrocybe sororia*	LE 24912	Russia	OM524387	OM523963
*Agrocybe* sp.	LE11405	Turkmenistan	JN684794	-
*Agrocybe subpediades*	LE 217898	Russia	JN684795	-
*Agrocybe subpediades*	LE 217899	Russia	JN684790	-
*Agrocybe vervacti*	GLM 45870	Germany	-	AY207143
*Agrocybe vervacti*	NL_2653	Hungary	-	MK277506
*Cyclocybe cylindracea*	ANGE318	Italy	KM260145	KM260150
*Cyclocybe cylindracea*	ANGE315	Italy	KM260144	KM260149
*Cyclocybe parasitica*	BRQ02/24	-	-	AY219580
*Deconica chionophila*	CBS 657.87	France	MH862112	MH873799
*Deconica chionophila*	CBS 659.87	France	MH862114	MH873801
*Deconica coprophila*	CBS 182.37	-	MH855878	MH867388
*Deconica coprophila*	CBS 181.37	-	MH855877	MH867387
*Deconica montana*	CBS 101791	Norway	MH862762	MH874362
*Deconica montana*	CBS 599.87	France	MH862108	MH873797
*Hebeloma affine*	NI270904	Canada	FJ436320	EF561632
*Hebeloma olympianum*	BK 21_Nov_98_0	-	-	AY038310
*Hebeloma velutipes*	PBM 2277	-	-	AY745703
*Hemistropharia albocrenulata*	T15	China	MH697851	MH697861
*Hemistropharia albocrenulata*	NL_5161	USA	-	MK278139
*Hypholoma capnoides*	GLM 45937	Germany	-	AY207211
*Hypholoma fasciculare*	TNS Kasuya B384	Japan	KC477654	KC603725
*Hypholoma fasciculare*	FO 46696	Germany	-	AF291340
*Hypholoma subericaeum*	GLM 45940	Germany	-	AY207215
*Hypholoma subericaeum*	H15	-	-	AF261629
*Kuehneromyces mutabilis*	CBS 205.32	Belgium	MH855288	MH866740
*Leratiomyces ceres*	G0672	Germany	-	MK278280
*Leratiomyces similis*	-	-	-	AF042009
*Leratiomyces squamosus*	PRM 922211	Czech Republic	MH043620	MH036179
*Meottomyces dissimulans*	MushroomObserver. Org_352303	USA	MW692353	MW692361
*Phaeogalera stagnina*	OS403	Norway	KC842390	KC842461
*Pholiota lenta*	TENN 074792	China	MN209742	MN251130
*Pholiota lenta*	TENN 074640	USA	MN209743	MN251131
*Pholiota lubrica*	TENN 074763	China	MN209749	MN251137
*Pholiota lubrica*	TENN 074784	China	MN209750	MN251138
*Pholiota spumosa*	TENN 074772	China	MN209776	MN251159
*Pholiota spumosa*	TENN 074778	China	MN209775	MN251158
*Protostropharia islandica*	MacroFF16013	Italy	KY914476	KY914475
*Protostropharia semiglobata*	v166	-	-	AF261625
*Psilocybe cubensis*	DNA 2052	-	KF830094	KF830083
*Psilocybe cyanescens*	CBS 10197	-	-	AF261620
*Psilocybe silvatica*	DAOM 187832	-	AY129362	AY129383
*Psilocybe stuntzii*	VT1263_D216	-	-	AF042567
*Pyrrhulomyces astragalinus*	TENN 074962	USA	MT187979	MT228845
*Pyrrhuylomyces amariceps*	TENN 071890	USA	MG735284	MN251114
*Pyrrhuylomyces amariceps*	PBM 2975	USA	HQ832448	HQ832462
*Stropharia aeruginosa*	S7	China	MH697848	MH697856
*Stropharia ambigua*	AFTOL_ID 726	Unknown	AY818350	AY646102
*Stropharia coronilla*	CBS 534.50	France	MH856747	MH868269
*Stropharia coronilla*	GLM 46028	Germany	-	AY207301
*Stropharia rugosoannulata*	S22	China	MH697846	MH697854

**Table 2 jof-09-00383-t002:** Vitamins and elements content of *A. striatipes* and *A. praecox* (g/100 g dry weight). Missing data are indicated by “-”.

Nutrient Content	*A. striatipes*	*A. praecox*
Protein (g/100 g)	5.66	2.05
Crude polysaccharide (g/100 g)	0.75	0.64
Ca (mg/kg)	78.5	33.4
Fe (mg/kg)	21.2	59.7
Zn (mg/100 g)	6.87	3.54
Mn (mg/100 g)	1.76	2.82
Se (mg/100 g)	0.048	0.23
Vitamin B1 (mg/100 g)	0.0850	0.0525
Vitamin B2 (mg/100 g)	0.333	0.123
Vitamin A (μg/100 g)	10	30
Vitamin C (mg/100 g)		2.0
Vitamin D (μg/100 g)	44.1	2
Vitamin E (mg/100 g)	-	0.12

**Table 3 jof-09-00383-t003:** Nutrient content of *A. striatipes* and other eight edible mushrooms (g/100g dry weight). Missing data are indicated by “-”.

Nutrient Content	*A.* *striatipes*	*A.* *praecox*	*C.* *cylindracea*	*C.* *salicacicola*	*P.* *placentodes*	*P.* *otreatus*	*L.* *edode*	*F.* *velutipes*	*A.* *bisporus*
ASP	0.42	0.12	1.65	2.31	2.63	1.33	1.43	0.78	1.97
THR *	0.23	0.075	0.99	1.25	1.81	0.83	0.95	0.66	1.1
SER	0.22	0.081	0.95	1.04	0.99	0.81	0.85	2.45	1.1
GLU	0.78	0.21	2.9	3.36	3.13	3.49	5.47	2.11	4.25
GLY	0.22	0.075	0.81	1.05	0.94	0.81	0.76	1.18	1.01
ALA	0.23	0.075	1.28	1.74	1.26	1.58	0.74	1.24	2.27
CYS	-	-	0.24	0.36	0.30	0.27	0.37	0.12	0.39
VAL *	0.21	0.074	0.96	1.05	0.95	0.9	1.08	0.77	1.09
MET *	0.052	0.18	0.29	0.32	0.16	0.27	0.49	0.12	0.4
ILE *	0.16	0.067	0.75	0.66	0.81	0.59	0.72	0.54	0.81
LEU *	0.26	0.087	1.42	0.90	1.43	1.15	1.06	0.9	1.69
TYR	0.11	0.043	0.53	0.37	0.59	0.46	0.23	0.83	0.58
PHE *	0.19	0.066	0.39	0.42	0.79	0.45	0.6	0.71	0.51
HIS	0.24	0.039	0.36	1.61	0.40	0.37	0.42	0.35	0.5
LYS *	0.27	-	0.86	2.71	0.85	0.77	1.06	1.09	1.08
ARG	0.30	0.093	0.9	1.95	1.06	0.73	0.91	0.58	1.12
PRO	0.24	0.062	1.07	1.03	0.87	0.88	0.57	0.64	2.22
GABA	0.017	-	-	-	-	-	-	-	-
T	4.14	1.447	16.4	22.13	19.24	15.7	17.71	15.07	22.1
E	1.37	0.649	5.66	6.41	7.07	4.96	5.96	4.79	6.68
NE	2.77	0.798	10.74	15.72	12.17	10.74	11.75	10.28	15.42
E/T	0.33	0.44	0.35	0.29	0.37	0.32	0.34	0.32	0.30
E/N	0.49	0.81	0.53	0.4	0.58	0.46	0.51	0.47	0.43

* T: Total amino acids; E: The total essential amino acids; N: The total non-essential amino acids. The asterisks refer to essential amino acids. Abbreviations: *A*., *Agrocybe*; *C*., *Cyclocybe*; *P*., *Pleurotus*; *L*., *Lentinus*; *F*., *Flammulina*; *A*., *Agaricus*.

**Table 4 jof-09-00383-t004:** The maximum allowable limits of heavy metals in edible mushrooms.

Heavy Metals	Category of Edible Mushrooms	Maximum Allowable Limits (mg/kg)	Content in Cultivated *A. striatipes*	National Food Safety Standard
Pb	Edible mushrooms and their products	≤0.5	0.0262	GB 2762-2022
Cd	Edible mushrooms and their products	≤0.2	0.047	GB 2762-2022
As	Edible mushrooms and their products	≤0.5	0.036	GB 2762-2022
Hg	Edible mushrooms and their products	≤0.1	0.053	GB 2762-2022

## Data Availability

All sequence data are available in NCBI GenBank following the accession numbers in the manuscript.

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
