# Peer review of "A New Species *Agrocybe striatipes*, also a Newly Commercially Cultivated Mushroom with Highly Nutritional and Healthy Values"

_jof, 2023, doi:10.3390/jof9030383_

Round 1
Reviewer 1 Report
Dear author,
The manuscript has some information deficit and is not well structured. Some information must be required for the manuscript
1. In the title what "heathy" refers to ?
2. As the title suggested "newly commercial cultivated mushroom " but the cultivation is missing in the manuscript
3. The mushroom specimen was collected from Xinwen County, but the latitude and longitude or other details such as the association and habitat of the collection is missing.
4. In the material methods, the protocols used for Nutritional composition analysis and mineral analysis are not mentioned.
4. Detail of the design in experiments are not mentioned.
5. The title of the manuscript indicates"highly nutritional and heathy values" but poorly written discussion on nutritional aspects
6. Table-1 can be included as a supplementary file
7. Cultivation part or substrate requirement (if not in patent process) can be added
8. Spell-check the manuscript thoroughly
Thank you
Reviewer 2 Report
The current manuscript entitled “A new species Agrocybe striatipes, also a newly commercial cultivated mushroom with highly nutritional and heathy values” by Li et al. reported a new species named Agrocybe striatipes from China. The results were supported by phenotypic and molecular phylogenetic studies using ITS and nrLSU genes. Furthermore, the mushroom was characterized by nutritional and elemental composition. After careful reading, this study is interesting, timely, novel, and suitable for publication in the JoF journal. I would like to suggest some minor improvements in the current version of the manuscript. My specific comments are:
1. Correct the grammatical error in the title.
2. Add geocoordinates of sampling locations in the materials section with a map depicting the actual location in China.
3. Line 82: Add mushroom before specimen.
4. Line 124-128: Add abbreviations for heavy metals (Pb, Cd, Hg, As). Also, provide brief methodology about sample digestion and instrument used (AAS or ICP?).
5. Add references for all protocols used in this study.
6. The maximum allowable limits of heavy metals in table 4 should be rechecked. Add a reference to WHO/FAO/other to support these limits. I think Cd is 0.10 and not 0.20.
7. The HPLC chromatogram in Figure 4 is unclear. Please indicate the meaning of different color lines (each chromatogram corresponds to a separate sample).
8. It is desirable to add a conclusion section and extend the discussion.
9. The captions of supplementary figures (FigA1-etc) are unclear. Analysis of samples of what?
Round 2
Reviewer 1 Report
The manuscript can be accepted
Author Response
Thank you very much for your very valuable comments concerning our article.
